# Engineered Liver-On-A-Chip Platform to Mimic Liver Functions and Its Biomedical Applications: A Review

**DOI:** 10.3390/mi10100676

**Published:** 2019-10-07

**Authors:** Jiu Deng, Wenbo Wei, Zongzheng Chen, Bingcheng Lin, Weijie Zhao, Yong Luo, Xiuli Zhang

**Affiliations:** 1State Key Laboratory of Fine Chemicals, Department of Chemical Engineering, Dalian University of Technology, Dalian 116024, China; dengjiu@mail.dlut.edu.cn (J.D.); wwb1004@hotmail.com (W.W.); zyzhao@dlut.edu.cn (W.Z.); yluo@dlut.edu.cn (Y.L.); 2Integrated Chinese and Western Medicine Postdoctoral research station, Jinan University, Guangzhou 510632, China; chenmond@foxmail.com; 3College of Pharmaceutical Science, Soochow University, Suzhou 215123, China

**Keywords:** liver-on-a-chip, drug hepatotoxicity, drug metabolism

## Abstract

Hepatology and drug development for liver diseases require in vitro liver models. Typical models include 2D planar primary hepatocytes, hepatocyte spheroids, hepatocyte organoids, and liver-on-a-chip. Liver-on-a-chip has emerged as the mainstream model for drug development because it recapitulates the liver microenvironment and has good assay robustness such as reproducibility. Liver-on-a-chip with human primary cells can potentially correlate clinical testing. Liver-on-a-chip can not only predict drug hepatotoxicity and drug metabolism, but also connect other artificial organs on the chip for a human-on-a-chip, which can reflect the overall effect of a drug. Engineering an effective liver-on-a-chip device requires knowledge of multiple disciplines including chemistry, fluidic mechanics, cell biology, electrics, and optics. This review first introduces the physiological microenvironments in the liver, especially the cell composition and its specialized roles, and then summarizes the strategies to build a liver-on-a-chip via microfluidic technologies and its biomedical applications. In addition, the latest advancements of liver-on-a-chip technologies are discussed, which serve as a basis for further liver-on-a-chip research.

## 1. Introduction

The liver is the largest intracorporeal organ in the human body and plays a predominant role in several pivotal functions to maintain normal physiological activities [1] such as blood sugar and ammonia level control, synthesis of various hormones, and detoxification of endogenous and exogenous substances [2]. Normally, the liver has a tremendous regenerative capacity to cope with physical and chemical damage. However, injury caused by adverse reactions to drugs (e.g., aristolochene and ibuprofen) and chronic diseases (e.g., viral and alcoholic hepatitis) may impair its ability to perform physiological functions [3,4].

Although in vivo models are commonly established in mammals to study liver functions, especially for pharmaceutical research, the accuracy of this kind of model is still unsatisfactory [5]. For example, roughly half of the drugs found to be responsible for liver injury during clinical trials did not result in any damage in animal models in vivo [6]. In addition, as a parenchymal organ, liver cells are continuously exposed to a variety of abundant exogenous substances. Moreover, it is inconvenient to observe highly dynamic biological processes in the current in vivo animal models.

Based on these facts, it is necessary to establish a reliable liver model in vitro for in-depth understanding of the physiological/pathological processes in the liver and the development of drugs for liver diseases. Currently, the liver models used for in vitro studies commonly include bioreactors (perfusion model of an isolated liver system) [7], 2D planar primary rat hepatocytes [8,9], 3D-printed liver tissue [10,11], liver organoids [12,13], and liver-on-a-chip systems [14,15,16]. To date, many previous reviews have discussed the differences in these models [17,18,19,20]. However, it is well known that liver-on-a-chip technology is innovative to manage liver microenvironments in vitro, and a variety of liver chips have emerged [18,20,21,22]. However, there is still no comprehensive review of the strategies to fabricate liver chips or their broad applications in various fields. The purpose of this review is to summarize the strategies to build liver-on-chips via microfluidic technologies and their applications.

We first introduce the physiological microenvironment of the liver, especially the cell composition and its specialized roles in the liver. We highlight the simulation objects of a liver-on-a-chip, including the liver sinusoid, liver lobule, and zonation in the lobule. Secondly, we discuss the general strategies to replicate human liver physiology and pathology ex vivo for liver-on-a-chip fabrication, such as liver chips based on layer-by-layer deposition. Third, we summarize the current applications and future direction. Finally, challenges and bottlenecks encountered to date will be presented.

## 2. Physiological Microenvironment of the Liver

### 2.1. Cell Types and Composition

The liver is composed of many types of primary resident cells such as hepatocytes (HCs), hepatic stellate cells (HSCs), Kupffer cells (KCs), and liver sinusoid endothelial cells (LSECs), which form complex signaling and metabolic environments. These cells perform liver functions directly and are connected to each other through autocrine and paracrine signaling. Below, we review each cell type and its contributions to liver functions along with their importance in the context of toxicity. The characteristics of each cell type are summarized in Table 1.

#### 2.1.1. Parenchymal Cells

Parenchymal cells, also called hepatocytes, are highly differentiated large epithelial cells (20–30 μm) responsible for the major liver functions [23] such as metabolism of blood sugar, decomposition of ammonia, and synthesis of bile acids. They comprise ~60% of total cells and ~80% of the total mass in the liver [24]. The main function of hepatocytes is metabolism of both internal and external substances. With a large number of mitochondria (1000–2000/cells), peroxisomes (400–700/cells), lysosomes (∼250/cells), Golgi complexes (∼50/cells), and endoplasmic reticulum both rough and smooth, each hepatocyte acts as a metabolism factory [23]. Nonetheless, the metabolic capacity of each hepatocyte is not exactly the same because of different oxygen pressures, nutrient supplies, and hormone concentrations in the various zones of liver lobules. Another feature of hepatocytes is polarization [25]. Physiologically, spatially adjacent hepatocytes are closely arranged in cords to form liver plates with strong polarization characteristics that allow substances to enter from the blood for excretion with bile.

Drug metabolism is the most studied function of hepatocytes, including both phase I and II, even though non-parenchymal support contributes to xenobiotic metabolism. Cytochrome P450 (CYP 450) family members, terminal oxygenases distributed on the endoplasmic reticulum and mitochondrial inner membrane, are the critical phase I metabolic enzymes in drug metabolism of the liver. They also have important effects on cytokines and thermoregulation. In particular, subtypes CYP 1A2, 2A6, 2C9, 2C19, 2D6, 2E1, and 3A4 are involved in almost all aspects of drug phase I metabolism, including oxidation, reduction, hydrolysis, and dehydrogenation. Phase II modifications are mostly carried out by cytosolic enzymes termed transferases that allow drug excretion by the kidneys after sulfation or glucuronidation.

#### 2.1.2. Hepatic Stellate Cells

HSCs, also called fat-storing cells, perisinusoidal cells, and lipocytes, reside within the space of Disse formed between hepatic cords and sinusoidal endothelial cells, which represent approximately 8% of hepatic cells. They have the same characteristics as fibroblasts and mainly play roles in vitamin A storage. Stellate cells play major roles in maintaining the morphology of LSECs and the progression of liver fibrosis. They exhibit various forms in different physiological environments, i.e., resting and activated states. Physiologically, stellate cells are in the resting state under normal conditions. However, when stellate cells are activated by changes in the microenvironment, such as alcohol intake and viral infections, the cells proliferate and migrate. The synthesis of collagen I and α-smooth muscle actin increases rapidly, leading to extensive extracellular matrix (ECM) deposition. Recent studies have shown that stellate cells are also involved in immune-mediated liver injury, causing secondary damage to the liver.

#### 2.1.3. Hepatic Sinusoidal Endothelial Cells

LSECs are long and slender endothelial cells in direct contact with liver blood flow [26,27]. They represent a major fraction of non-parenchymal cells (~48%) with extended processes. In addition, LSECs express SE-1 and CD31 proteins abundantly and thus can be identified easily. Unlike vascular endothelial cells, LSECs not only acts as a physical barrier for blood circulation, they also have their own characteristics by lacking a basement membrane and rich in fenestrations [27]. However, these features are lost when a lesion occurs. LSECs express endothelial nitric oxide synthase (eNOS) protein, which are affected by blood flow shear and vascular endothelial growth factor (VEGF), thus adapting to changes in blood flow velocity and pressure in liver sinusoids. LSECs are also involved in the immune response of the liver, such as phagocytosis of particles and adhesion of immune cells. In addition, LSECs express toll-like receptors that detect exogenous substances and self-apoptotic products that trigger the inflammation pathway.

#### 2.1.4. Kupffer Cells

Kupffer cells are macrophages that reside in the liver, accounting for approximately 80%–90% of all fixed macrophages in the body and about 15% of total liver cells [26]. They are predominantly localized in the lumen of hepatic sinusoids and are anchored to the surface of LSECs by long extended processes. KCs are irregular in shape and about 10–13 μm in size. The main function of KCs is to remove particulates and foreign matter from the portal vein by phagocytosis. In addition, KCs are closely related to homeostasis of the liver environment. They release many cytokines, such as TNF-α, IL-1, and IL-6, which are involved in immunomodulation. For example, tumor-necrosis factor alpha (TNF-α) released by KCs acts on LSECs, leading to fibrin deposition in liver tissue, which may cause ischemia and hypoxia [28,29]. Recent studies have found that KCs also participate in antigen presentation [30].

#### 2.1.5. Biliary Epithelial Cells

Biliary epithelial cells are the main epithelial cells located in the bile duct wall with a diameter of about 10 μm. They are one of the few cell types rarely studied in the liver because of their small effect on liver functions. However, recent studies have shown that the bile excretion pathway of drugs is related to hepatotoxicity [31]. Moreover, these cells express multiple bile receptors, which may be of interest to study cholestasis-induced liver disease.

#### 2.1.6. Other Non-Parenchymal Cells

In addition to the above five kinds of cells, there are many other kinds of cells in the liver, such as neutrophils, natural killer (NK) cells, and infiltrating macrophages [24]. Physiologically, the numbers of such cells are small, but when the liver develops inflammation, these cells enter the liver rapidly in large amounts, which should be considered under pathological conditions. The main function of these cells is to release a large number of cytokines and chemokines to regulate the liver. Increasing evidence has revealed that these cells are closely related to immune-mediated hepatotoxicity [32].

### 2.2. Simulation Objects of a Liver-on-a-Chip

#### 2.2.1. Liver Sinusoid

Liver sinusoid, a lacuna between adjacent liver plates (Figure 1C), is the physiological microenvironment of the liver with strong permeability to exchange materials between liver cells and blood flow [33]. In addition to containing the main liver cell types, liver sinusoid has its own specific structure in which HCs and LSECs are separated by the sinusoidal space, and hepatic stellate cells and extracellular matrix fill the gaps between HCs and LSECs. KCs are not attached to the lumen of blood vessels. The vertices of HCs fuse with each other to form bile canaliculi. The characteristics of liver sinusoids are such that cells in the liver can be approximately seen as assembled layer-by-layer. There are a large number of liver sinusoid models in vitro to reconstruct the physiologically relevant and controlled environment of liver. To date, the most used strategies employ additional polycarbonate (PC) membranes or layering by a laminar.

#### 2.2.2. Liver Lobule

The liver lobule, which is considered as the smallest functional unit of the liver, appears as a polygon of approximately 1.1 mm in diameter and 1.7 mm in length [23]. Each lobule consists of hepatocytes radiating from the central vein and are separated by vascular endothelial cells. There are about 1 × 10^3^ lobules in each human liver. At the center of each hexagon, there is a large vein called the central vein. The corners of the hexagon contain three conduits, the hepatic portal vein, hepatic artery, and bile duct. They are characterized by blood concentrating from six corners to the center, while bile moves from the center to the outside. Cell capturing using traps and micropatterning methods are the conventional methods to reconstruct hepatic lobules. Recently, 3D printing has become another rapid and simple method [35].

#### 2.2.3. Zonation in the Lobule

Liver zonation is an evolutionary optimized segregation of the broad liver functions into spatial, temporarily defined, and highly specialized zones [36]. In liver zonation, different pathways are carried out in different zones—even in single cells. As shown in Figure 1C, cells in the periphery of liver lobules are relatively rich in oxygen and glucose, resulting in relatively higher albumin and urea synthesis. In contrast, internal cells possess relatively higher glycolysis than cells in peripheral zones. Liver zoning also leads to differences in hepatotoxicity. As shown in Figure 1D, because of the lower CYP activity in zone 1, less cells are damaged, while the oxygen content in zone 2 is lower and CYP activity is therefore enhanced, thereby showing greater hepatotoxicity [34]. Such heterogeneity and functional plasticity of the liver are survival strategies for each cell to perform simultaneously without affecting each other and to use resources efficiently.

## 3. General Strategies for in Vitro Liver Models

As shown in Figure 2, the currently used in vitro liver models commonly include 2D planar primary hepatocytes [8,9], bioreactors (perfusion model of an isolated liver system) [7], 3D-printed liver tissue [37,38], 3D liver spheroids [39,40], and liver-on-a-chip. Table 2 compares the advantages and disadvantages of each model.

Because of the convenience and ease of handling, the conventional 2D culture of hepatocytes has been widely used as an in vitro liver model to study drug metabolism and cytotoxicity. However, most 2D-cultured hepatocytes lose their intrinsic biochemical cues and cell-cell communications necessary to maintain the physiological phenotype and cannot fully recapitulate liver-specific functions [34]. The perfusion model of an isolated liver system employs blood filtration, which is used to assist treatment of liver dysfunction and related diseases, and rarely used as a liver model to study pathology and physiology in vitro. Rapid development of 3D printing technology has provided a promising approach for in vitro liver models, which precisely controls the placement of cells, allowing the formation of separate hepatocyte and nonparenchymal cell (NPC) compartments. However, defects of the bulky dimension without flow mobility make it difficult to use for rapid, high throughput drug and toxicology evaluations. Furthermore, current printing accuracy cannot always allow placement of individual cells, which makes it impossible to reproduce physiological cues faithfully. 

Primary hepatocyte aggregation culture forms 3D spheroids as a representative liver model to mimic early liver development. Many studies have shown that culturing hepatocytes within a 3D ECM-like matrix not only mimics the architecture of the liver, but also improves cell-to-cell and cell-to-matrix interactions and supports intrinsic liver functions, including production of albumin and urea as well as phase I and II drug metabolism [41]. Construction of liver organoids as a model system is an appealing experimental approach to exploit the physiological mechanisms that occur during organ development and regeneration [39]. The main techniques to generate organoids are the formation of spheroids by aggregation of cells and extracellular matrix components [40]. Such 3D liver organoid structures can meet the needs of the pharmacological and toxicological industry for drug screening [42]. The disadvantage of spheroids is that the cells are distributed randomly without formation of spatial organization i.e., liver spheroids neither possess the typical hepatic cord-like alignment of polarized hepatocytes nor sinusoids lined with endothelial cells reflecting the in vivo condition [17]. Furthermore, the sizes of spheroids are difficult to unify, and necrosis can occur in the center of large spheroids. 

Recently, microfluidics-based cell culture devices have gained the most interest for biological and biomedical applications. The hepatocytes cultured within these devices under flow conditions allow for more frequent nutrient and waste exchange compared with conventional models. In addition, the hepatocytes better recapitulate liver-specific functions. For example, gradients of oxygen/hormones can be created to model zonal liver phenotypes. However, liver-on-chip fabrication requires many trivial operations, which will be described in detail in Aection 4.

In vitro liver models are critical for hepatology studies and drug development for liver diseases. An important aspect of these models is the cell source. There are three major types of cells to reconstruct liver tissue in vitro: primary human hepatocytes (PHHs), hepatic-derived cell lines, and stem cell-derived hepatocytes [43]. Table 3 compares the advantages and disadvantages of each cell source in detail. PHHs are considered as the gold standard of liver models in vitro. Isolated PHHs exhibit many intrinsic liver characteristics, including phase I and II metabolic enzyme activities, glucose metabolism, and ammonia detoxification. However, culturing PHHs on dishes has several issues such as lose of liver-specific functions, unsuitability for long-term studies, high costs, and donor variation.

Hepatic cell lines, such as HepG2 (derived from human hepatocellular carcinoma of a 15-year-old male) and HepaRG (terminally differentiated hepatic cells derived from a hepatic progenitor cell line of a female hepatocarcinoma), have been widely used in toxicological investigations, because they have a stable phenotype, are essential for drug metabolism and toxicity responses, and are easily manipulated with unlimited proliferation [59,60]. For example, HepG2 cells have been used for toxicological and pharmacological research since the 1970s. However, compared with PHHs, the cells lines cannot represent the phenotype of in vivo hepatocytes and their drug reactions are inaccurate because of the low activities of CYP450 and transporters such as organic anion transporting polypeptide and sodium taurocholate co-transporting polypeptide [18]. Therefore, hepatic-derived cell lines are only suitable for the early stages of drug safety and screening assessment.

Stem cell-derived hepatocytes have become a new alternative liver cell source [61]. Du et al. employed induced pluripotent stem cells (iPSCs) differentiated into hepatocytes and endothelial cells, and then encapsulated them in fibers to form liver tissue-like constructs [51]. In addition, the 3D-aggregated stem cells can be differentiated into liver organoids with stable functions including albumin secretion, liver-specific gene expression, urea production, and metabolic activities [33,34,62]. However, manipulation of stem cells is not as simple as that of cell lines and requires specific induction factors during a >15-day culture period to obtain differentiated liver cells. Moreover, hepatocytes differentiated from iPSCs show less albumin secretion, exhibit dramatically reduced CYP450 activity, and express immature markers such as alpha-feto protein. Therefore, the maturation of organoids is hardly comparable with that of organoids formed by PHHs [18,63,64].

It has been demonstrated that culturing hepatocytes with non-parenchymal cells increases functionality and longevity. In addition, coculture provides multiple types cellular interactions and recovers cellular polarity, which more similar to in vivo conditions. Moreover, coculture of multiple types of cells forms many typical liver structures such as sinusoid [17], lobule [31], and biliary systems [34].

## 4. Liver-On-A-Chip Technology

Mimicking the liver in vitro remains a great challenge. Even coculture systems hardly simulate the complexity of the liver, because different types of cells are mixed and seeded randomly in coculture, which cannot form the complex cellular architecture or manipulate cell-to-cell interactions. However, rapid development of microfabrication and microfluidic technologies has provided a promising approach to establish microscale functional liver constructs on a chip. Moreover, microfluidic devices have many attractive advantages compared with conventional culture. For example, a microfluidic device can easily generate a concentration gradient, control cellular spatial distribution, and provide a flow environment. Therefore, researchers have developed various strategies to build a liver-on-a-chip via microfluidic technology for biological and biomedical applications. Table 4 summarizes the current general methods to build liver-on-chips and their advantages and disadvantages. 

### 4.1. Liver Chips Based on 2D Planar Culture

Monolayer culture, also called 2D planar culture, has been widely used in the early stages of drug screening because it is easy to handle and amenable to screening large numbers of compounds in a short amount of time. However, mounting evidence indicates that 2D planar culture of hepatocytes results in rapid loss of hepatic marker expression and phenotypes within hours and is unsuitable for long-term culture. Moreover, cell-cell communication between hepatocytes and non-parenchymal cells improves liver functions, whereas these interactions are weak in 2D planar culture. To mimic the complexity of interactions of each cell type of the liver, researchers have developed methods for patterning and coculturing hepatocytes with other cell types on 2D substrates. Micropatterned substrates not only alter the distribution of different types of cells in a controllable manner, but also provide suitable biochemical cues for both parenchymal and non-parenchymal cells. For example, Ho et al. designed hepatic lobule arrays to pattern and coculture HepG2 cells and human umbilical vein endothelial cells (HUVECs) [65]. By manipulating dielectrophoresis, the original randomly distributed cells in the microfluidic chamber were able to form the desired pattern, which mimicked the morphology of a lobule. In addition, they found that the activity of the CYP450 enzyme was 80% higher compared with non-patterned HegG2 cells after two days of culture. However, the surface characteristics of 2D materials, such as stiffness and hydrophilicity, also affect the phenotype and function of hepatocytes. Therefore, the development of suitable materials has become the direction of 2D planar culture.

### 4.2. Liver Chips Based on Matrixless 3D Spheroid Culture

Aggregating hepatocytes into a 3D spheroid is another conventional method and promising in vitro model for hepatic metabolism and cytotoxicity research. Hanging drop technology facilitates cell aggregation, resulting in spheroids [53,66]. Aeby et al. used hanging hydrogel drops to form primary human liver microtissues for more than nine days [67]. In addition, Boos et al. combined primary human liver microtissues with embryoid bodies in the same hanging drop platform and found that the metabolites of primary human liver microtissues were directly transported to the EBs [68].

The cell-repellent plate method is commonly used to self-assemble liver cell spheroids [59,60,61,69]. Desai et al. used magnetic nanoparticles to modify PHHs and then applied magnetic force to rapidly assemble and easily handle the spheroids [59]. Moreover, in toxicological applications, by integrating microsensors with a liver-on-a-chip, the metabolic parameters and cell viability of a single spheroid could be monitored without microscopy [60]. Furthermore, the established human liver spheroid model could be used to study other liver diseases. For example, 3D spheroids of cocultured HepaRG and HSCs enabled testing of drug-induced liver fibrosis in vitro. After applying drugs, these cocultured spheroids presented HSC-specific gene expression and fibrotic features, including HC activation, and collagen secretion and deposition [69]. In addition to static culture, Tostoes et al. showed that human hepatocyte spheroids maintained liver-specific protein synthesis, CYP450 activity, and phase II and III drug-metabolizing enzyme gene expression and activity in a perfusion bioreactor system for two to four weeks [41].

Alternatively, a microwell array can also be applied to generate 3D spheroids in a high-throughput and controllable manner [33,51]. For example, Miyamoto et al. showed that hundreds of uniform HepG2 spheroids were generated with a TASCL (tapered stencil for cluster culture) device [51]. The size of the spheroids relied on the size of the microwell. In addition, the spheroids showed high viability and an albumin secretion activity. Moreover, the liver chip could be fabricated with a perfusion chamber that provided a fluidic shear stress biomimetic microenvironment. Subsequently, Ma et al. designed a reversible concave microwell-based polydimethylsiloxane (PDMS)-membrane-PDMS sandwich multilayer chip [33]. Their results showed that 1080 HepG2/C3A cell spheroids could be perfused in parallel in long-term culture on the chip, which significantly improved the establishment of cell polarity and enhanced liver-specific functions and metabolic activities.

Inspired by lattice growth mechanisms in materials science, Weng et al. designed a liver chip with a hexagonal culture chamber to mimic physiology and control the assembly of liver cells into an organotypic hierarchy [34]. First, they deposited primary liver cells (PLCs) onto a micropatterned hydrophobic PDMS membrane. The cell-coated membrane was then enclosed within the hexagonal culture chamber. The medium was introduced by flow in the chamber from each corner of the hexagonal chamber, which simulated the portal vein function. They found that the PLCs formed a scaffold-free hierarchical tissue and represented complex functional dynamics at the tissue level, such as dose effects of acetaminophen-induced hepatotoxicity. 

### 4.3. Liver Chips Based on Matrix-Dependent 3D Culture

In liver tissue engineering, an ECM-like scaffold is required to facilitate cell adhesion, support cell growth, and enhance cell-matrix interactions. Therefore, various ECM components (natural and synthetic) have been applied to liver chips to improve their liver functions for pharmaceutical and cytotoxicity applications. Researchers have used an ECM within a liver chip to maintain and mimic the native microenvironment. For example, Toh et al. developed a multiplexed microfluidic 3D hepatocyte chip for in vitro drug toxicity testing [62]. The chip was coated with collagen, a natural ECM component, to support hepatocyte adhesion and growth (Figure 3a). They found that hepatocytes cultured in the coated chip showed both cell-cell and cell-ECM interactions, and maintained hepatocyte synthetic and metabolic functions. 

Hepatocytes have also been introduced into liver chips with a pre-gel ECM component solution [54,70,72]. Hegde et al. demonstrated a method to culture hepatocytes in a collagen sandwich configuration in which hepatocytes were immobilized at the bottom chamber and the top chamber remained open for perfusion, as shown in Figure 3b [70]. Their results demonstrated that hepatocytes within the chip exhibited not only higher albumin and urea secretion, but also higher collagen secretion under perfusion. Interestingly, hepatocytes showed a well-connected cellular network with bile canaliculus formation over two weeks of perfusion culture. Similarly, Jang et al. also showed that Matrigel-embedded HepG2 cells formed a bile canaliculus structure over 14 days of perfusion culture [72]. Moreover, they reported HepG2 cell cultivation with an indirect flow but without a physical barrier. These data demonstrated that a microfluidic chip improves and stabilizes hepatic functions by mimicking an in vivo hepatocyte environment. Furthermore, Bavli et al. extended liver chip applications for real-time monitoring of mitochondrial respiration by co-encapsulating oxygen-sensing beads in collagen [54]. The HepG2/C3A cells formed 3D aggregates in the microfluidic chip, which provided native-like physiological shear forces and a stable oxygen gradient. These results showed that their system permitted detection of minute shifts from oxidative phosphorylation to glycolysis or glutaminolysis, which demonstrated the unique advantage of organ-on-chip technology.

Hydrogels have various intrinsic critical features to mimic native mechanical and structural cues that promote cell adhesion, proliferation, and differentiation. For example, Christoffersson et al. developed a modular and flexible hyaluronan and poly(ethylene glycol) (HA-PEG) hydrogel in which the mechanical properties and hydrogelation kinetics could be conveniently tuned by modulating the degree of crosslinking and temperature, respectively [73]. HepG2 cells or iPSC-derived hepatocytes (hiPS-HEPs) were encapsulated in the HA-PEG hydrogel in a perfusion device to enable implementation to a liver-on-a-chip. In addition, they compared the HA-PEG hydrogel with agarose and alginate. HepG2 cells encapsulated within all hydrogels formed spheroids with high viability, and the albumin and urea secretion were the highest in alginate hydrogels. Furthermore, hiPS-HEPs migrated and grew in 3D within Arg-Gly-Asp (RGD) peptide-modified HA-PEG hydrogels and showed increased viability and higher albumin secretion compared with other hydrogels. Zhu et al. synthesized state-specific liver microtumors using thermal-sensitive hydrogels with tailored stiffness and 3D scaffolds [74]. Their results indicated a close relationship between tissue biomechanics and drug efficacy, which provided a powerful tool for discovery and optimization of tissue-specific stroma-reprogrammed combinatorial therapy.

The bottom-up tissue engineering approach creates relatively bionic 3D tissues containing multiple types of hierarchically assembled cells. 3D liver spheroids can be generated easily by various methods, as described above. However, the major limitation of spherical morphology is a uniform supply of oxygen and nutrients, where cells located at the center are prone to die or lose their functions because of the hypoxic environment. Therefore, researchers have developed a cell-laden microfiber strategy for effective delivery of oxygen and nutrients via molecular diffusion. Yajima et al. packed HepG2 cells into the core of sandwich-type anisotropic microfibers. In addition, vascular endothelial cells were seeded on the fiber surface to form vascular network-like conduits between fibers. These cell-laden microfibers were further cultivated in a perfusion chamber. HepG2 cells within the microfiber not only exhibited high viability and functions, but also mimicked the structure of the hepatic lobule and sinusoidal in vitro [75]. 

Although hydrogel scaffolds resemble both structural and mechanical cues of the ECM, the artificial 3D systems still lack the appropriate native growth factors that promote cell growth and sustained cell functions. Therefore, a decellularized liver matrix (DLM) has become the most promising candidate for engineering native-like liver tissue. Lu et al. developed a biomimetic 3D liver tumor-on-a-chip with a DLM and gelatin metharcyloyl (GelMA) in a microfluidic 3D dynamic cell culture system (Figure 3c) [75]. As expected, the liver-on-a-chip integrated with DLM-GelMA better recapitulated the tumor microenvironment, such as essential scaffold proteins, growth factors, stiffness, and shear stress. Moreover, it demonstrated dose-dependent responses to the toxicity of acetaminophen and sorafenib.

### 4.4. Liver Chips Based on Layer-by-Layer Deposition

As shown in Figure 1C, the liver sinusoid, a functional repetitive microvascular unit that is formed by the sinusoidal wall composed of endothelial cells connected to the portal vein and hepatic artery, has unique structural characteristics. In detail, HCs and LSECs are separated by the hepatic sinusoidal space, HSCs and extracellular matrix fill the gap, and mainly KCs are free in the vascular lumen. This layered structure of each type of cell allows building liver-on-chips by layer-by-layer deposition. Therefore, researchers have developed a layer-by-layer cell-coating technique to create in vitro 3D vascularized tissue. For example, Sasaki et al. coated cells with layer-by-layer nanofilms of fibronectin and gelatin, and the coated cells reconstructed homogenous, dense, well-vascularized liver tissue with high a cellular function (albumin production) and cytochrome P450 activity [76]. Ahmed et al. designed an approach to seed hepatocytes, endothelial cells, and stellate cells sequentially on hollow fiber membranes [44]. The cells attached on the fiber surface, self-assembled, and formed tissue-like structures around and between fibers, which synthesized albumin and urea for 28 days. 

Owing to the flexibility of microfabrication and the microfluidic technique, a multichannel microfluidic chip has become a promising platform to recapitulate the critical features of the liver sinusoid. Mi et al. constructed a liver sinusoid based on the laminar flow on a chip [77]. Specifically, HepG2 cell- and HUVEC-laden collagens were synchronously injected into the microfluidic chip. Then, by taking advantage of laminar flow, the two collagen layers formed a clear borderline, and the HUVECs in the collagen self-assembled into a monolayer by controlling cell density and injection of a growth factor. 

Kang et al. reported a dual channel microfluidic platform, where primary hepatocytes and endothelial cells were cocultured within the channel to mimic the architecture of the liver sinusoid [45]. Hepatocytes maintained their normal morphology under continuous perfusion and produced urea for at least 30 days. In addition, the sinusoid-on-a-chip could be used to analyze replication of the hepatitis B virus. Rennert et al. established a liver organoid integrating all major types of cells (hepatocytes, stellate cells, endothelial cells, and macrophages) in a perfused biochip that was used a porous membrane to mimic the space of Disse. Endothelial cells and macrophages were seeded on the top side of the membrane, and hepatocyte-like HepaRG and stellate cells were cocultured on the opposite side of the membrane [78]. The liver organoid displayed clear differentiation and structural reorganization. Moreover, perfusion increased hepatobiliary secretion of 5(6)-carboxy-2′,7′-dichlorofluorescein and enhanced hepatocyte microvillus formation. More evidence demonstrated that the artificial liver sinusoid maintained high viability and in vivo-like morphology in long-term perfusion culture. In addition, the flow rate was not only related to albumin and urea responses, but also enhanced HGF production and CYP450 metabolism [79,80,81]. Furthermore, Deng et al. used all cell lines (HepG2, LX-2, EAhy926, and U937 cells) to build a liver sinusoid-on-a-chip with artificial liver blood flow and biliary efflux flowing in the opposite direction (Figure 4a). The all cell line liver chip was used to test the hepatoxicity of acetaminophen with other drugs. The results were similar to the “gold standard” primary hepatocyte plate model, indicating that the all cell line liver chip provided an alternative approach to investigate drug hepatoxicity and drug-drug interactions [31]. 

Another advantage of an organ-on-a-chip is that it can achieve real-time monitoring by integrating with sensors. As shown in Figure 4b, Moya et al. used an inkjet-printed oxygen sensor with a liver-on-a-chip to evaluate metabolic activity in real-time [82].

### 4.5. Liver Chips Based on 3D Bioprinting

Rapid development of 3D printing technology has provided a promising approach for liver tissue engineering, which facilitates automated and high-throughput fabrication of precisely controlled 3D architectures. 3D bioprinting can produce anatomically accurate liver anatomy including the specific spatial structure and vascular network of the liver. The unique aspects of 3D bioprinting techniques are making them increasingly popular tools to manufacture in vitro liver models to study liver diseases and screen drugs. For example, Noroma et al. bioprinted human liver constructs comprising primary hepatocytes, hepatic stellate cells, and endothelial cells to model methotrexate- and thioacetamide-induced liver injury leading to fibrosis [83]. After exposure to these compounds, liver injury was detected, including hepatocellular damage, and deposition and accumulation of fibrillar collagens, which indicated that the 3D-bioprinted liver recapitulated compound-induced liver injury responses. Similarly, Nguyen et al. found that 3D-bioprinted liver tissue not only effectively modeled drug-induced liver injury, but also distinguished between highly related compounds with differential profiles [84].

Even though 3D-bioprinted liver tissue has been considered as an ideal liver model in vitro, a printed 3D liver analogue in a conventional plate or transwell cannot provide the dynamic perfusion condition. To overcome this limitation, a bioprinted liver was further integrated with microfluidic and perfusion devices to form a liver-on-a-chip. For example, HepG2/C3A spheroids were printed with GelMA ink and perfusion cultured in a bioreactor chamber for 30 days. The printed cell spheroid presented liver-specific functions including secretion of albumin, alpha-1, antitrypsin, transferrin, and ceruloplasmin [86]. Massa et al. developed a perfusable vascularized 3D liver construct via a sacrificial bioprinting technique [92]. Agarose fibers were printed in per-polymerized GelMA, which encapsulated HepG2/C3A cells. After GelMA polymerization, the agarose fibers were removed and HUVECs were seeded within the hollow channel to form a vascularized construct. The encapsulated HepG2/C3A cells exhibited high viability within the vascularized construct, demonstrating a protective role of the introduced endothelial cell layer. In another study, Grix et al. printed a complex cell-laden lobule with a hollow channel system using a stereolithographic bioprinting approach [88]. This printed liver organoid also showed higher albumin and cytochrome P450 expression compared with the monolayer control over 14 days of cultivation. Furthermore, Lee et al. generated a 3D liver-on-a-chip with multiple cell types using decellularized ECM bio ink and integrated it with a microfluidic device containing vascular and biliary fluidic channels [10]. Their results demonstrated that the liver functionalities were significantly enhanced by the formation of the biliary system on a chip, which becomes an effective potential candidate for drug discovery.

### 4.6. Liver Chips Based on Other Technologies

In addition to the above techniques, other methods can be applied to the manufacture of liver-on-chips, such as cell microarrays, microwell systems, and hanging drops [90]. The most notable feature of cell microarrays and microwell systems is the high throughput for large-scale screening of drugs at the early stage. Micropillars and traps are common methods used to form a cell microarray. For example, Lee et al. [89] designed a microchip platform for 3D culture of Hep3B human hepatic cells. This device features 532 micropillars and corresponding microwells that combine for high-throughput assessment of compound hepatotoxicity. The device was demonstrated to be suitable for 3D cell encapsulation, gene expression, and rapid toxicity assessment. The hanging drop method is another approach for liver chip fabrication. Frey et al. [68] reported a microfluidic hanging drop platform that seamlessly integrated liver metabolism into an embryonic stem cell test. This device was operated by gravity-driven flow to ensure constant inter-tissue communication as well as rapid and efficient exchange of metabolites.

## 5. Applications of a Liver-on-a-Chip

In the past decades, liver-on-chips have been a useful tool for drug screening and toxicity testing, prediction of metabolism, establishment of liver disease models, and studying the interactions of multiple organs. Because a microfluidic approach aims at mimicking the physiological and pathological conditions, various liver chips have been described for 2D and 3D culture of hepatocytes alone or in coculture with other types of cells in the liver for long-term culture and to study toxicity, metabolism, and disease. Table 5 summarizes the typical applications of liver-on-a-chip systems.

### 5.1. Drug Screening and Toxicity Testing

Drug-induced liver injury remains a significant source of clinical attrition, restrictive drug labeling, and post-market withdrawal of therapeutics [93,94]. Because of the ability to mimic in vivo physiological parameters, organ-on-a-chip devices enhance hepatocyte functions to assess the cellular behavior responses to drugs, which offer an alternative to animal experimentation [94]. Toh and colleagues developed multichannel microfluidics with a 3D engineered microenvironment to maintain the synthetic and metabolic functions of hepatocytes [62]. The multiplexed channels allow simultaneous management of drug doses at a concentration gradient to hepatocytes, enabling prediction of hepatotoxicity in vitro. Yu and colleagues designed a perfusion-incubator-liver-chip for 3D spheroid culture with a tangential flow, which maintained cell viability for over 24 days [95]. Then, chronic drug responses to repeated dosing of diclofenac and acetaminophen were evaluated by this device. Based on the channel modification technique, a 3D liver sinusoid-mimicking model was established [55]. Three hepatotoxins were evaluated by this liver chip. IC50 values were closely aligned with the LD50 values in mice, thus demonstrating the hepatotoxicity testing effectiveness of the proposed liver model. Except for endpoint assays to assess drug toxicity, dynamic information can be used as well to assess a drug’s mechanism of action. Bavli and colleges constructed a liver-on-chip device capable of being maintained for more than a month in vitro under physiological conditions to allow real-time analysis of minute shifts from oxidative phosphorylation to anaerobic glycolysis in the process of mitochondrial stress [54]. Through this microfluidic platform, the dynamics and strategies of cellular adaptation to mitochondrial damage induced by rotenone and troglitazone were revealed. 

A liver-on-chip provides a credible tool for studying relatively complete drug action pathways. Prot and colleagues integrated transcriptomic, proteomic and metabolomics profiles by cultivating liver cells in microfluidic biochips treated with or without acetaminophen (APAP), which allowed a more complete reconstruction of APAP-induced injury pathways. In addition, the validity of the results was confirmed by comparisons with in vivo studies [96,97]. However, other than the liver, drug hepatotoxicity metabolism is also associated with other organs. Thus, an organ-on-chip makes it possible to study the interactions between multiple organs. A series of multiple organ chips have been established to mimic the physiological environmental systems of multiple organs, such as intestines with the liver [97], nephridium with the liver, and lungs with the liver [98]. 

### 5.2. Prediction of Metabolism 

A goal of a liver-on-chip is to establish a hepatocyte microenvironment in vitro consistent with that in vivo. Compared with cell culture techniques, the comprehensive metabolic capacity of liver cells cultured on organ chips is enhanced greatly. For example, primary hepatocytes hosted in 3D heparin-coated microtrenches secreted high levels of albumin and urea for over four weeks [55]. In addition, a liver-on-chip provides a dynamic flow environment for hepatocytes to perform higher albumin synthesis and urea excretion (detoxification) compared with static cultures [79]. Precision-cut liver slices can also be incubated directly on a microfluidic-based biochip, and the metabolic rate was significantly improved by embedding slices in Matrigel-based microfluidic chips [99]. Another important function of the liver is participation in drug metabolism. A liver-organ-chip can be used to evaluate phase I and II metabolisms in the liver [79] and study the first pass metabolism of drugs by integrating a gut-like structure in the front end of the liver chip [100], which is difficult to reproduce in vitro by conventional cell culture systems. Zhou et al. developed a five-chamber microsystem—two for coculturing hepatocytes with HSCs and three other chambers integrating aptamer-modified electrodes to monitor secretion of transforming growth factor-β [101]. This microsystem is capable of monitoring paracrine crosstalk between two cell types communicating via signaling molecules.

In conclusion, liver-on-chips based on microfabrication technology make it feasible to establish a liver model closer to the actual physiological environment in vivo, which is characterized by coculturing multiple types of cells under physiological flow conditions, high metabolic activity of hepatocytes, and establishment of complex and reliable cellular microenvironments. 

### 5.3. Establishment of Liver Disease Models

Based on the characteristic of organ-on-chips controlling the external microenvironment of cells, a series of models for liver diseases—including alcoholic liver disease [103], fatty liver disease, liver fibrosis, acute liver injury, and patient-specific liver diseases—were established in the form of a liver-on-chip.

In Lee’s study, rat primary hepatocytes and HSCs were cocultured in a fluid activity chip to observe structural changes, which exhibited a decrease in hepatic functions with the increase in ethanol concentration [47]. To develop a human in vitro model of non-alcoholic fatty liver disease, HepG2 cells [104] and primary hepatocytes [105] were cultured under 2D and 3D perfused dynamic conditions with free fatty acid supplementation, respectively. The models allowed for sustained culture of hepatocytes in vitro, which were used to investigate FFA-induced intrahepatic triglyceride accumulation (steatosis), which initially leads to a benign condition but can progress to more advanced conditions of steatohepatitis and fibrosis. Primary hepatocytes and LSECs, which were isolated from control and cirrhotic humans, were cocultured on the chip to mimic the in vivo physiological sinusoidal environment [81]. A human, 3D, four-cell, sequentially layered, and self-assembled microfluidic liver model demonstrated the development and characteristics of early fibrotic activation induced by 30 nM methotrexate as indicated by the expression of alpha-smooth muscle actin and collagen, and increased stellate cell migration [106].

A liver-on-a-chip device has also been used to study host/pathogen interactions. A 3D microfluidic liver culture system was constructed to provide a valuable preclinical platform for hepatitis B virus (HBV) research, which is capable of recapitulating all steps of the HBV life cycle, including the replication of patient-derived HBV and maintenance of HBV cccDNA [52]. The pathological process may reflect the specificity and genetics of the patient. Therefore, the principle of disease therapy based on other non-specific models cannot be applied to everyone. In these examples, a liver-on-chip provides an advanced model that better preserves the liver phenotype and can employ different cell types critical to facilitate the development of personalized/targeted medicine. Primary hepatocytes and LSECs, which were isolated from control and cirrhotic humans, have also been cocultured on a chip to mimic the in vivo-specific physiological and pathology of the sinusoidal environment [81]. Schepers and colleges described perfusable liver chip-cultured 3D organoids consisting of primary and iPSC-derived cells from a patient of interest. In their study, organoids were encapsulated and cultured in C-trap architecture for at least 28 days [58]. This strategy can be applied to other microfluidic organ models, which provides an opportunity to query patient-specific liver responses in vitro. Another hiPSC-induced liver organoid-on-a-chip system also demonstrated a promising organoid-based liver chip platform with applications in precision medicine and disease modeling [107].

### 5.4. Fabrication of Multiple Organs on a Chip

As an organ, the liver does not perform its function alone in vivo, but undertakes higher physiological functions together with other organs or tissues. An organ-on-chip allows for crosstalk between multiple organs to be studied by connecting a liver chip with organotypic models. Another important application of liver-on-chips is the combination with other organ chips, called “body-on-a-chip” or “human-on-a-chip”, used to study complex mechanisms in disease and drug screening. The most common organ-organ interaction studies are based on the intestines and liver, which are physically closest to each other.

Lee and colleagues built a gut-liver coculture chip with a PK model to predict first-pass metabolism by comparing the PK profile of paracetamol obtained by this chip with the known profile in humans. The clearance of the drug in this chip was significantly slow, and the gap was closed by improving the absorption surface area and metabolic capacity of the chip [97]. In addition, the gut-liver chip mimicked the absorption and accumulation of fatty acids in the gut and liver. Moreover, the effects of TNF-α, butyrate, and α-lipoic on hepatic steatosis via different mechanisms were evaluated by this chip [108]. To evaluate nanoparticle interactions with human tissues, a gastrointestinal tract and liver tissue system was constructed by combining the human intestinal epithelium and liver represented by coculture of enterocytes (Caco-2) with mucin-producing cells (TH29-MTX) and HepG2/C3A cells. High doses of nanoparticles induced aspartate aminotransferase release, indicating liver cell injury. Therefore, this device successfully simulated the uptake, metabolism, and toxicity of acetaminophen in vitro [109] (Figure 5A).

In addition to the intestines, the liver is associated with tumors, lungs, and skin, and can be used to study how substances are metabolized in multiple organs or tissues. For example, a liver and tumor-combined organ-on-chip was used as a PK-PD model to interpret drug actions in multiple organs [110]. The anti-cancer activity of luteolin was evaluated in this study, which was significantly weaker than that in 2D culture. These results revealed that simultaneous metabolism and tumor-killing actions likely resulted in a decreased anti-cancer effect. This study demonstrates that multiple organs on a chip established by combining the liver with a tumor is a useful tool for gaining insights into the mechanisms of drugs by interactions among multiple organs. Moreover, a lung/liver-on-a-chip has been reported (Figure 5B). Liver spheroids were connected in a single circuit, and normal human bronchial epithelial cells were cultured at the air-liquid interface. Aflatoxin B1 (AFB1) toxicity in lung tissues decreased when liver spheroids were present in the same chip circuit, indicating that the liver-mediated detoxification protected lung tissues [98]. The lung/liver-on-a-chip platform presented here offers new opportunities to study the toxicity of inhaled aerosols or to demonstrate the safety and efficacy of new drug candidates targeting the human lung. A simplified liver-kidney-on-chip model has been reported, which was used to investigate the biotransformation and toxicity of aflatoxin B1 (AFB1) and benzoalphapyrene (BαP) [111]. Coculture of human artificial liver microtissues and skin biopsies in multi-organ-chip was performed to emulate the systemic organ complexity of the human body, and each tissue had 1/100,000 of the biomass of their original human organ counterparts. After 14 days of coculture in a fluid flow environment, crosstalk between the liver and skin tissues was observed. This model facilitates exposure of skin at the air-liquid interface and provides a potential new tool for systemic substance testing [112].

In addition to interactions between the liver and a single organ, the crosstalk between the liver and more than one organ/tissue has been studied by connecting multiple organs in one microfluidic system. A homeostatic long-term coculture of human liver equivalents with either a reconstructed human intestinal barrier model or human skin biopsy platform has been reported. This platform provides pulsatile fluid flow within physiological ranges at low medium-to-tissue ratios and supports submersed cultivation of an intact intestinal barrier model and an air-liquid interface for the skin model during their coculture with liver equivalents [102]. Moreover, a four-organ-chip was employed to evaluate systemic absorption and metabolism of drugs in the small intestines, as shown in Figure 5C. Metabolism by the liver and excretion by the kidney are key determinants of the efficacy and safety of therapeutic candidates. Within two to four days, establishment of reproducible homeostasis among the co-cultures appeared at fluid-to-tissue ratios near to those in the physiological environment, and this homeostasis was sustained for at least 28 days [113].

These results clearly support the importance of advanced interconnected multi-organs in microfluidic devices for application to in vitro toxicity testing as well as optimized tissue culture systems for in vitro drug screening.

## 6. Challenges and Future Directions

The microfluidic technique offers substantial benefits to generate a functional liver on a chip. Accumulating evidence shows that liver-on-a-chip technology has achieved significant success in biomanufacturing for recreating key aspects of the liver, drug development in hepatotoxicity testing, and investigation of fundamental mechanisms in liver disease. However, the development of liver-on-a-chip technology is still in the early stage and many challenges remain. A critical issue is whether the results from liver-on-chips can replace those from animal experiments. To resolve this issue, the following points are worthy of attention.

As the building material of liver chips, a sustainable and reliable liver cell source is one of the key limitations. Primary hepatocytes and multiple types of non-parenchymal cells—rather than animal cells—are the best candidates, even though dedifferentiation hepatocytes in vitro remain to be overcome. Recently, mutable human embryonic stem cells and iPSCs have been considered as the most promising alternative sources. In particular, liver organoids have shown a similar spatial organization as the liver, which are able to reproduce some of the functions of the liver. However, to generate and control physiologically relevant structural, mechanical and biochemical cues that instruct directional differentiation remain great challenges. In addition, even though recent studies have begun to simulate bile tubes and shown hepatocyte polarization, most existing liver models cannot allow for the bile canaliculus and bile acid secretion, which may have a significant effect on liver functions and subsequent applications.

Another critical technical challenge arises from the stability of readouts and excessive complexity of operations. There have been a variety of liver chip devices to date, but none of them have received FDA approval, which is mainly due to the lack of uniform testing standards. The readouts of recent liver chips are often end point results, lacking real-time dynamic monitoring, which ignores many important physiological processes. Moreover, the throughput of liver chips is relatively low, which is unsuitable for the rapid high throughput of industrial applications. Biosensors integration will make up for the drawbacks of liver chip devices in terms of readouts and throughput. It is foreseeable that future organ chips will integrate many biosensors to meet the requirements for the automatization and monitorization. In addition, the use of high-tech detection technologies, such as live cell imaging and super resolution microscopy, are beneficial as well to the future commercial applications of liver chips.

Finally, a liver-on-a-chip itself cannot recapitulate the communication between different organs, thereby lacking the pharmacokinetic properties and toxic effects between other organs during hepatotoxicity testing. The immune system also plays a critical role in liver infection, disease progression, and drug-induced hepatotoxicity. To our knowledge, there is still no liver chip containing the immune system. In addition, to achieve crosstalk of a liver-on-a-chip with other organs, studies are anticipated to develop the next generation of multi-organs-on-chips such as a liver-kidney chip, liver-intestine chip, liver-immune chip, and finally a human-on-a-chip. In the near future, with the development of microengineering and microfluidic technologies, there may emerge a large number of fast, high-throughput liver-on-a-chip devices and liver organoids that can be widely used in pathological studies and environmental toxicology.

## Figures and Tables

**Figure 1 micromachines-10-00676-f001:**
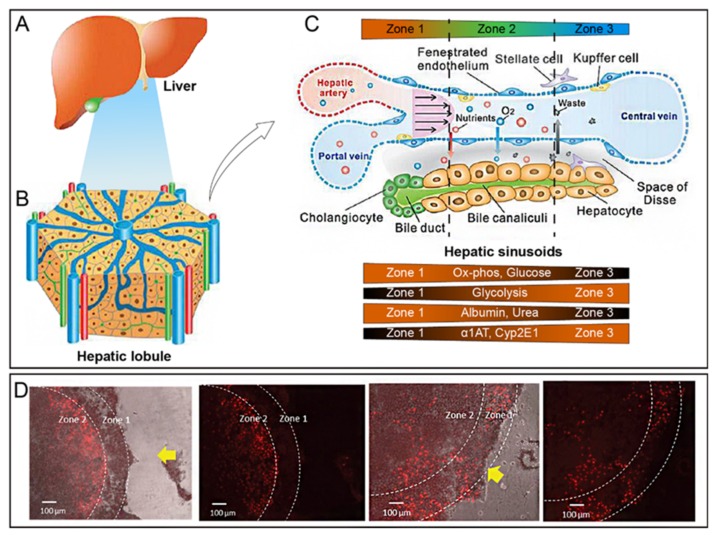
Cellular composition and anatomical microstructure of the liver. (**A**) Shape of the liver. It is a red-brown V-shaped organ divided into right and left parts by the hepatic artery, portal vein, hepatic vein, and bile ducts. (**B**) The liver lobule has a hexagonal shape with a diameter of about 1 mm and thickness of about 2 mm. (**C**) Zonation in the lobule. Reproduced with permission from [33]. (**D**) Zonal heterogeneity of acetaminophen-induced hepatotoxicity. The yellow arrow indicates the flow direction. Reproduced with permission from [34].

**Figure 2 micromachines-10-00676-f002:**
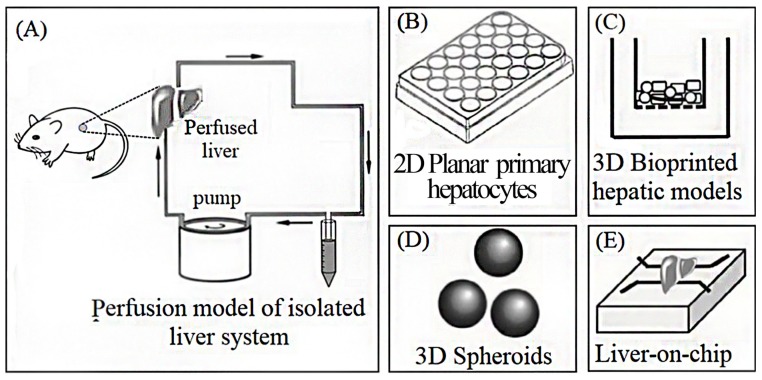
Liver models used commonly in vitro. (**A**) Perfusion model of an isolated liver system; (**B**) 2D planar primary rat hepatocytes; (**C**) 3D-printed liver tissue; (**D**) 3D spheroids; (**E**) liver-on-chip.

**Figure 3 micromachines-10-00676-f003:**
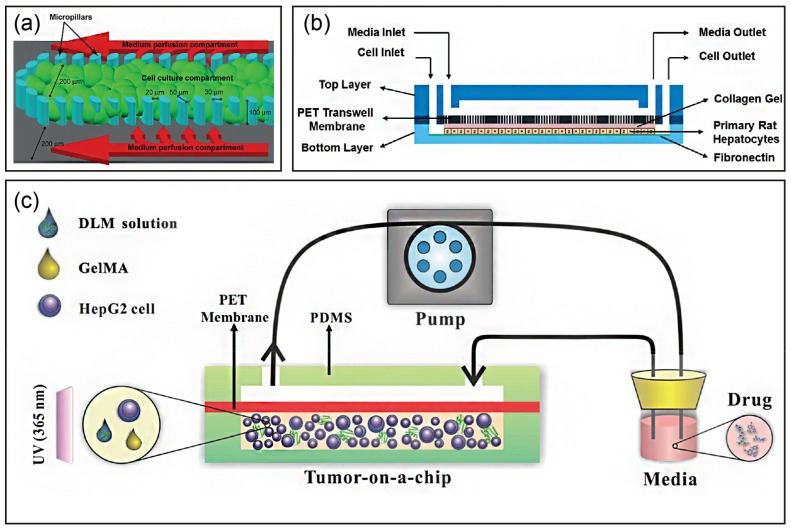
Strategy to build a liver-on-a-chip in matrix-dependent 3D culture: (**a**) liver cell culture channel of the multiplexed cell culture chip. Reproduced with permission from [62]. (**b**) Cross-sectional view of the assembled device showing that hepatocytes in collagen gel are introduced and cultured in the bottom layer and growth medium is introduced through the top layer. Reproduced with permission from [70]. (**c**) Schematic of HepG2-laden decellularized liver matrix with gelatin methacryloyl (DLM-GelMA) in a microfluidic device. Hydrogel precursors are injected into a microfluidic device using a pipette and photopolymerized by UV exposure to form HepG2-laden DLM-GelMA for subsequent drug screening. Reproduced with permission from [71].

**Figure 4 micromachines-10-00676-f004:**
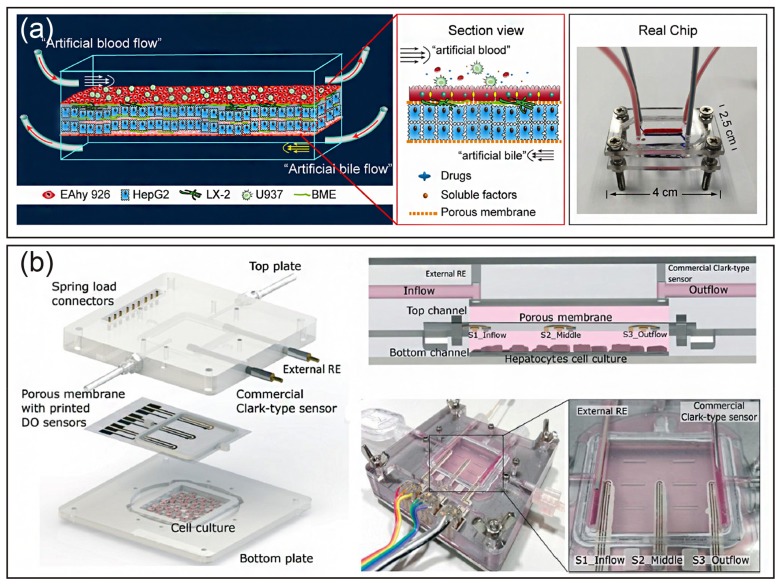
Establishment of a liver-on-a-chip using layer-by-layer deposition. (**a**) Schematic of the liver sinusoid structure and LSOC microdevice. Reproduced with permission from [31]. (**b**) Schematic, cross-section, and real image of an oxygen sensor-integrated liver chip. Reproduced with permission from [82].

**Figure 5 micromachines-10-00676-f005:**
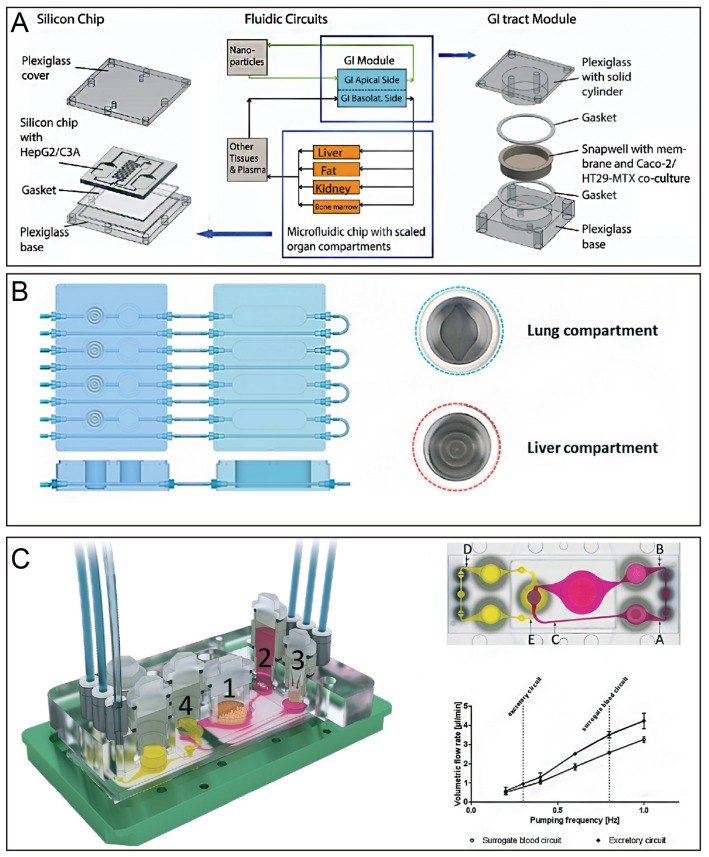
(**A**) Gastrointestinal tract and liver tissue system construct by combining the human intestinal epithelium and liver represented by coculture of enterocytes (Caco-2) with mucin-producing cells (TH29-MTX) and HepG2/C3A cells. Reproduced with permission from [109]. (**B**) Lung/liver-on-a-chip, in which liver spheroids were connected in a single circuit and normal human bronchial epithelial cells were cultured at the air-liquid interface. Reproduced with permission from [98]. (**C**) The microfluidic four-organ-chip device. (i) 3D view of the device comprising two polycarbonate cover plates, a PDMS-glass chip accommodating a surrogate blood flow circuit (pink), and an excretory flow circuit (yellow). Numbers represent the four tissue culture compartments for the intestines (1), liver (2), skin (3), and kidney (4). (ii) Central cross-section of each tissue culture compartment aligned along the interconnecting microchannel. (iii) Average volumetric flow rate plotted against the pumping frequency of the flow circuit. Reproduced with permission from [113].

**Table 1 micromachines-10-00676-t001:** Main cell types of the liver and their features.

Cell	Type	Diameter (μm)	Proportion (number)	Features
Parenchymal	-	-	-	-
hepatocytes	Epithelial	20–30	60%–65%	Large in size, abundant glycogen, mostly double nuclei.
Non-parenchymal	-	-	-	-
Kupffer cells	Macrophages	10–13	~15%	Irregularly shaped, mobile cells, secretion of mediators.
liver sinusoid endothelial cells	Epithelial	6.5–11	16%	SE-1, CD31, fenestrations, none basement membrane.
hepatic stellate cells	Fibroblastic	10.7–11.5	8%	Vitamin-storing,
Biliary Epithelial Cells	Epithelial	~10	Little	Distinct basement membrane. Containing unique proteoglycans, adhesion glycoproteins.

**Table 2 micromachines-10-00676-t002:** Advantages and limitations of in vitro liver models (note: these methods may have crossovers).

In Vitro Approaches	References	Advantages	Limitations
Monolayer	[8,9]	Easily manipulated, low-cost, good repeatability.	Cannot recapitulate in-vivo like cellular morphology and 3D microenvironment, loss of cell polarity, poor function.
Co-culture	[44,45,46]	Multi-cellular environment, cell-cell interaction, improve functions and longevity, cellular polarity.	Difficult isolation of NPCs, variations of NPCs, differentiation status and viability are varied depending on culture conditions.
3D culture	[11,47,48,49,50]	Recapitulation of 3D microenvironment and ECM properties, improve gene and protein expression, improve functions and longevity, cellular polarity.	Complicated methods of culture. Necrotic regions within 3D cellular models caused by oxygen diffusion.
Spheroids	[41,46,51]	In vivo-like microenvironment, cellular interaction, maintain liver-specific functionality over long term culture, enhanced CYP 450 and transporter expression, formation of secondary structure (e.g., bile canalicular-like structure).	Spheroid size limitation (~200 μm) and variations, necrotic cores, Oxygen and nutrient diffusion through cellular aggregates.
Liver-on-a-chip	[35,52,53,54,55,56,57,58]	Dynamic microenvironment, suitable for co-culture, 3D culture, and spheroid, improve liver-specific, functionality, enhanced CYP 450 and transporter expression formation of secondary structure, pattern cells spatially, high through put and low cost.	Complicated methods of operate chip and culture cell in the chip, required perfusion systems, non-specific binding of drugs to chip materials, may wash away molecules in the chamber under perfusion, no standard yet.

**Table 3 micromachines-10-00676-t003:** Advantages and limitations of cells used in liver-on-chips.

Cell Type	Advantages	Limitations
Primary hepatocytes (human, rat)	Liver intrinsic characteristics, including phase I and II metabolic enzyme activity, glucose metabolism, ammonia detoxification	Losing liver specific function; unsuitable for long-term; high cost; donor variation, difficult isolation
Hepatic-derived cell lines (HepG2, HepaRG, C3A)	Unlimited lifespan; easily manipulated; stable phenotype; essential for drug metabolism and toxicity response.	Drug reaction are inaccurate; low metabolic competence and rapid loss of expression of liver-specific enzymes/transporters.
Stem cell induces hepatocytes	A stable source of hepatocytes; liver organoid; stable functions including albumin secretion, liver-specific gene expression, urea production and metabolic activity.	Hardly manipulated; required specific induce factor; high cost; insufficient maturate.

**Table 4 micromachines-10-00676-t004:** Summary of the strategies used for liver-on-chip fabrication.

Strategies	References	Characteristics	Culture Period	Advantages	Disadvantages
Liver chip based on 2D planar culture	[65]	Pattern or capture hepatocytes in 2D form; co-culture with non-parenchymal cells.	Short term	Relatively easy and fast; suitable for high throughput screening.	No polarization; low cell-cell communication; depended on the nature of substrate.
Liver chip based on matrixless 3D spheroid culture	[33,41,51,59,60,61]	Hepatocytes form spheroid spontaneously, due to gravity or modification of material surface; also suitable for co-culture.	Medium to long term	Scaffold-free; easy to achieve mass production of uniform size; good part form for stem cell differentiation	Needs special technology, such as cell-repellent plate and hanging drop technique.
Liver chip based on matrix-dependent 3D culture	**[54,62,70,71,72,73,74,75]**	Encapsulate cells within a three-dimensional (3D) matrix, such as hydrogel, BME and collagen, which replicates the supportive functions of the extracellular matrix.	Long term	Provide support and fixation for cells; enhanced cell-cell and cell-matrix interaction; conducive to cell adhesion and regulate dynamic cue of cells	Dependent on matrix, such as stability, stiffness; batch-to-batch variability; potential immunogenicity and presence of biological contaminants; unpredictable effects on signaling pathways.
Liver chip based on layer-by-layer deposition	[33,44,45,76,80,85]	Pattern hepatocytes and nonparenchymal cells lay by lay by porous membrane or 3D printing technology, etc.	Long term	Easy to control the position of cell layers to mimic the distribution of liver cells; forming tightly connected endotheliocytes for perfusion; hepatocyte polarization and angiogenesis	Not suitable for organs with unclear cell stratification; depends on other auxiliary tool, such as membrane and bio-ink.
Liver chip based on 3D bioprinting	**[10,11,84,86,87,88]**	Cells and extracellular matrix are laid out according to a preset path through a 3D printer in the form of additive manufacturing.	Long term	Easy to construct complex 3D biological microscale structures with various cell types and biomaterials; time save and high throughput	Limited by printing accuracy, it is difficult to control individual cells; the properties of printed materials are not optimized enough.
Liver chip-based cell microarrays such as microwell systems	**[74,89,90]**	Seed cells in an array of well plates.	Medium to long term	High throughput; miniaturize and parallelize.	Lack of spatial distribution and cellular interactions of cells in vivo.
Liver chip-based hanging drops	**[68,91]**	Form 3D micro-tissues of cells (one type or multi-types) by hanging cells in drop.	Medium term	Controllable and reproducible spheroid formation; no need to use scaffold; each drop served as a culture compartment for a single microtissue that was suitable for high throughput screening.	Not suitable for long-term culture for chronic toxicity and chronic liver disease.

**Table 5 micromachines-10-00676-t005:** Typical applications of liver-on-a-chip systems.

Application	Reference	Cells Used	Description	Experimental Specifications
Drug screening and toxicity testing	**[95]**	Primary rat hepatocytes	A perfusion-incubator-liver-chip (PIC) was designed for 3D rat hepatocyte spheroids culture; chronic drug response to repeated dosing of Diclofenac and Acetaminophen were evaluated in PIC.	PIC system structure, functionality and optimization; Maintenance of cell function in PIC; application of PIC-cultured hepatocytes in drug safety testing.
Prediction of metabolism	**[100]**	Caco-2; HepG2	A microfluidic chip consists of two separate layers for Caco-2 and HepG2 was designed; first pass metabolism of a flavonoid, apigenin was evaluated as a model compound.	Gut-liver chip design for cells proliferation and differentiation; Paracellular permeability of intestinal barrier; first pass metabolism of apigenin.
Establishment of liver disease models	**[52]**	HepDE19; cryopreserved PHH; HepG2	A 3D microfluidic PHH system permissive to HBV infection; This system enables the recapitulation of all steps of the HBV life cycle, replication of patient-derived HBV and the maintenance of HBV cccDNA.	HBV patient-derived viruses and infections; exogenous stimulation of KC suppresses HBV replication.
Fabrication of multi-organ on a chip	**[102]**	HepaRG; human primary hepatic stellate cells; prepuce	A system for the co-culture of human 3D liver spheroids with human gut barrier and skin toward systemic repeated dose substance testing.	Fourteen-day performance of liver-intestinal co-cultures; 14-day performance of liver-skin co-cultures; repeated dose substance exposure.

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
