# Peer review of "Engineered Liver-On-A-Chip Platform to Mimic Liver Functions and Its Biomedical Applications: A Review"

_micromachines, 2019, doi:10.3390/mi10100676_

Round 1
Reviewer 1 Report
I have no further comments.
Author Response
Thank you for your comments, your recognition is the driving force of our work.
Sincerely.
Reviewer 2 Report
The manuscript has been written well and many aspects of the field of liver-on-a-chip has been covered.
If they expand future trend a little bit more, that would be great.
Author Response
Thank you for your comments. your recognition is the driving force of our work.
In the revised version (line 673-683), we highlight the importance of biosensors integration and the use of newly detection technologies for the automatization and monitorization of liver-on-chips for the future commercial applications.
Reviewer 3 Report
Engineered liver-on-a-chip platform to mimic liver functions and its biomedical applications: A review
The presented manuscript aims to review the state-of-the-art of liver-on-a-chip platforms designed for biomedical applications. The authors first summarize the physiological microenvironment of the liver, cell types, biological roles and composition, followed by a broad perspective of the standard strategies to build a liver-on-a-chip via microfluidic technologies and its biomedical applications. The review is well organized and focused in important aspects for liver-on-a-chip research. However, the importance of the integration of biosensors for the automatization and monitorization of these organs is missing. This is a very important aspect that need to be addressed in the near future for the progress and commercialization of these advanced microfluidic devices. I suggest that a new sub-section on this thematic should be added, or at least some paragraphs in section 6 (Challenges and future directions).
Overall, I think the manuscript is an interesting contribution to the readers of Micromachines journal and should be considered for publication after this minor improvement.
Author Response
Thank you for your valuable comments. your suggestion is significant for guiding the future direction of organ-on-a-chip technology. In section 6, we expanded paragraph 3 (line 673-683) to describe the drawbacks of readout and throughput of recent liver chips. we highlight the importance of the use of biosensors and newly detect technologies for the future industrial application. Hope to meet with approval.
This manuscript is a resubmission of an earlier submission. The following is a list of the peer review reports and author responses from that submission.
Round 1
Reviewer 1 Report
This paper has been built to provide a critical review on the existing Liver-on-a-Chip technology and their applications. Though a lot of research are being conducted in Organ-on-Chip technology, a review paper on these topics is essential for the community to collectively process the work going on around, and plan on how to move the technology forward. In that way, the chosen topic on Liver-on-Chips is highly relevant. The manuscript has also been well written which makes for an easy read.
However, in my opinion, a review paper is not just a conglomeration of the literature that is already available. For this review paper, one would have expected the authors to have conducted an in-depth and elaborate discussion on some of the challenges with Liver-on-Chip technology in terms of materials, fabrication, scalability, handling , etc. The authors do not bring in any critical analysis nor add any novel insight to this technology. Whatever is detailed in the manuscript is all well known among the people of the working in this community of Organ-on-Chip. This is a severe drawback of this work. The review thus seems incomplete. The authors have also not included the citation of some key works in Organ-on-Chip technology (such as works from Don Ingber, or Milica Radisic) which could be good reference points for building the review paper. Going by the discussion in Lines 622/662, this paper seems to have been presented more like a literature survey for a proposed student thesis on multiple-organ-on-chip, rather than as a review article. Therefore, I feel that this paper in its present form is not acceptable for publication at this time.
Reviewer 2 Report
The review and referred material in this manuscript for Liver-on-Chip research and development is comprehensive with updating information. The manuscript organization is reasonable. My concerns/comments are as follows.
Two corresponding authors are listed. Please confirm this. I don't check in details. Most figures shown in this manuscript do not come from the authors of this manuscript. I did not find any copyright permission mentioned through this manuscript. To get copy permission for these figures from original journals and original authors might be needed to avoid copyright/legal issue for MDPI publication.Reviewer 3 Report
The review entitled “Liver on a Chip” by Deng, et al., describes the development of in vitro liver models for pharmaceutical development applications. The authors cover much ground, leading to a high-level description of what is a very large and complex field with critical real-world applications. In doing so, their review does not provide depth required for utility to researchers nor is a clear novel perspective or conclusion provided that would help guide the field’s development. I applaud the authors for the heroic effort, but I strongly suggest they revise this review with a more specific focus or perspective that would enable them to provide sufficient depth through thoughtful review of existing work, particularly the last 5 years, and provide researchers in the field a useful reference. Additionally, the manuscript excludes non-liver models and many reviews of multi-organ systems and pharmacokinetic modeling/analysis that demonstrated critical concepts that are essential for any organ chip system, missing an opportunity to further provide a potentially highly-cited review. Given this assessment, I recommend this manuscript be rejected, however a substantial refocusing and revision, coupled with strong English language editing would make this of great interest to the community.
Major points:
There is not much depth to this review (e.g., 2D liver chip models are described using one citation) since the authors attempt to review nearly every aspect of in vitro liver model development. Given the massive extent of this research landscape, the authors should consider focusing onto a single sub-area for this review. The manuscript requires extensive editing for grammar and syntax by a native English speaker. Many areas are difficult to understand or incomprehensible. It is unclear what the main purpose of this review is as the authors provide a high-level summary of various efforts that do not provide insights or perspectives beyond recent reviews (e.g., https://www.sciencedirect.com/science/article/pii/S2352345X1730173X or https://pubs.rsc.org/en/content/articlelanding/2015/lc/c5lc00611b/unauth#!divAbstract). The work of major leaders in the field of microphysiological systems and organs chips (e.g., Ingber, Shuler, Marx, Bhatia, Griffith) is barely mentioned or not at all, which inherently misses many of the key principles and concepts required for a review of in vitro liver cultures. 1 and lines 43-52: this section should be expanded to clearly describe and review the listed in vitro liver methods (technically, perfused liver should be called “ex vivo” and perhaps even omitted from in-depth discussion entirely. The current review of the methods is too vague to do justice to the work in the field and the commercial developments/applications using some of these models. The authors also miss discussing another category of liver models: patterned liver cultures that do not require 3D printing or organ chip culture (e.g., cell microarrays, hanging drops, microwell systems). Additionally, it would make more sense to describe the hardware approaches (Fig. 1) AFTER describing normal liver morphology, cell types, and functions (Fig. 2) as the authors do in subsequent paragraphs. It would better lead to their eventual goal of describing Liver on Chip systems. Lines 58-61: please provide citations and discuss in greater depth. Lines 218-241: this is a critical point, and while the authors do call out the lack of accuracy in using cell line models, they do not provide references not adequate detail for why these models fail. This is true of primary cell and iPS cell culture approaches as well. This section also lacks depth concerning the relevance of cell co-cultures, which are as important as the cell source. Lastly, the point of inter-donor variability is raised, which is presented as a negative. While the phenomenon is true, the variability of donors reflects the variability of patients that will receive drugs and therefore should be incorporated into pharma development, including (especially) in vitro models given their grater scalability than animal models. Given these topic’s importance, please provide strong citation support and expand your assessments. Line 243: the concept of microenvironment is brought up multiple times, and the authors should make this more concise throughout the manuscript while also clearly differentiating between the physical and biological microenvironments during discussion as the current description merges the two at times while sometimes treating them separately.